# SIMILARITY-CONSTRAINED REWEIGHTING FOR COMPLEX QUERY ANSWERING ON KNOWLEDGE GRAPHS

## ABSTRACT

Machine learning models for answering complex queries on knowledge graphs estimate the likelihood of answers that are not reachable via direct traversal. Prior work in this area has focused on structured queries whose constraints are expressed in first-order logic. Recent work has proposed to extend such logical constraints with *soft entity constraints*, which require the answer to a query (also known as the *target variable*) to be similar or dissimilar to specified sets of entities.

A natural but unexplored generalization of this extension is to allow specifying similarity constraints not only on the answer to a query, but also on the values assigned to *intermediate variables*, which frequently occur in complex queries. In this work, we study this more general formulation and introduce SCORE: a computationally efficient and interpretable method for incorporating similarity constraints at arbitrary positions of a query. Unlike approaches that rely on deep neural networks, SCORE is based on a lightweight and interpretable score adjustment function that only requires tuning two parameters on a validation set.

Our experiments on a challenging benchmark over three different knowledge graphs demonstrate that on the special case of constraints on the target variable, SCORE is able to incorporate preferences without degrading overall query answering performance, with significantly increased ranking performance over a learned neural baseline. Moreover, SCORE maintains its performance in the more general setting with constraints on intermediate variables. Our code is available at https://anonymous.4open.science/r/score-simcqa.

## 1 INTRODUCTION

Knowledge graphs (KGs) have emerged as a fundamental data structure for organizing and reasoning about real-world information, encoding millions of entities and their relationships across diverse domains including encyclopedic knowledge (Bollacker et al., 2008), biomedical data (Himmelstein et al., 2017), and scientific literature (Ji et al., 2022). Their structured representation enables reasoning tasks that go beyond simple fact retrieval, supporting complex queries that involve multiple logical operations such as conjunction, disjunction, and existential quantification. These capabilities have made KGs relevant in applications such as question answering and recommendation systems (Zhou et al., 2020), drug discovery (Daza et al., 2023), and knowledge-based reasoning (Hogan et al., 2021).

However, knowledge graphs are inherently incomplete, missing a substantial fraction of true facts (Nickel et al., 2016). This incompleteness poses significant challenges for traditional symbolic query processing, which can only retrieve answers that are explicitly present in the graph. To address this limitation, machine learning approaches for *complex query answering* have been developed (Hamilton et al., 2018; Ren et al., 2020; Ren & Leskovec, 2020; Ren et al., 2024), which learn to estimate the likelihood of potential answers by reasoning over incomplete knowledge graphs.

Despite these advances, current complex query answering systems are limited to expressing constraints through first-order logic, which restricts their ability to capture nuanced preferences and domain-specific requirements. Recent work has begun to address this limitation by introducing *soft entity constraints* (Daza et al., 2025), which allows specifying that query answers should be similar or dissimilar to given sets of entities. For instance, when querying for "drugs treating diseases that

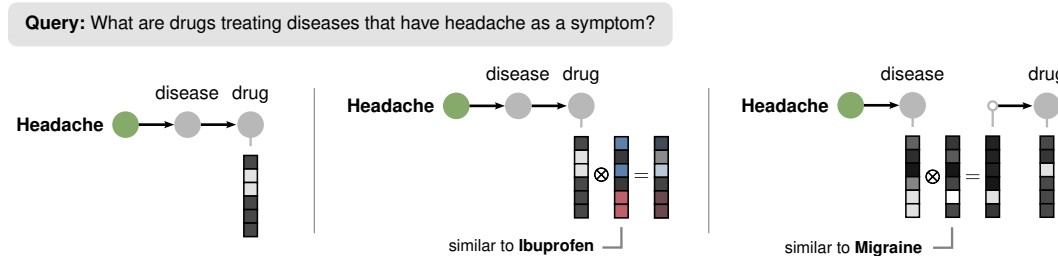

Figure 1: Illustration of three approaches for query answering on knowledge graphs, with vectors indicating entity scores at different steps. **Left**: a traditional query with purely logical constraints. **Middle**: incorporating similarity constraints in NQR (Daza et al., 2025), which supports only the target variable via uncalibrated score adjustments (shown in red and blue). **Right**: SCORE, which supports similarity constraints on arbitrary variables through normalized similarity scores.

have headache as a symptom," a user might specify preferences such as "drugs similar to Ibuprofen". While this formulation represents a significant step towards more expressive interfaces for querying a KG, it only supports similarity constraints on the final answers being sought, also known as the *target variable*. This limitation limits expressiveness, as constraints can also involve intermediate variables that appear during query processing. For example, one might specify constraints not only on the drug (target variable) but also on the diseases (the intermediate variable), such as "diseases like migraine".

When constraints apply only to target variables, similarity-based score adjustments can be applied as a post-processing step after the query has been evaluated. However, incorporating constraints on intermediate variables is not trivial, as they must be *integrated* into the query processing itself. Crucially, this integration depends on operating with *normalized scores*, rather than uncalibrated similarity scores, so that they can be meaningfully combined with other probabilistic components of the query answering model. Such an integration must preserve the logical structure of the query while incorporating similarity constraints at arbitrary positions, which is not supported by the methods introduced by Daza et al. (2025).

In this work, we address this fundamental limitation by studying the problem of *Similarity-constrained Complex Query Answering* (SimCQA) in its full generality, where similarity constraints can be specified on **any variable** within a complex query. We then propose SCORE (**S**imilarity-**CO**nstrained **RE**weighting), a computationally efficient method that extends existing neural query answering approaches to support similarity constraints at arbitrary query positions.

The key innovation of SCORE lies in its lightweight and interpretable score adjustment mechanism, which uses pre-trained entity embeddings to compute normalized similarity scores and integrates them into the query processing pipeline, as illustrated in Figure 1 on the right. Unlike approaches that require expensive end-to-end neural optimization, SCORE introduces only two hyperparameters that can be efficiently tuned on a validation set. Our contributions are summarized as follows:

- We formalize the problem of similarity-constrained complex query answering with constraints on arbitrary variables, generalizing previous work.
- We introduce SCORE, a lightweight and interpretable method for incorporating similarity constraints at any position in complex queries, requiring minimal computational overhead and hyperparameter tuning.
- We conduct comprehensive experiments on three challenging knowledge graph benchmarks (FB15K-237, NELL-995, and Hetionet), demonstrating that SCORE achieves substantial improvements in ranking quality and constraint satisfaction (NDCG@10 increases of 3–9 percentage points, representing 10–45% relative improvements) with respect to a neural baseline, while maintaining competitive performance on unconstrained queries.

## 2 RELATED WORK

**Complex query answering.** Neural approaches to complex query answering (CQA) have evolved from simple link prediction to multi-hop logical queries Ren et al. (2024). Early knowledge graph

embedding methods focused on predicting individual missing edges (Nickel et al., 2011; Bordes et al., 2013; Trouillon et al., 2016; Lacroix et al., 2018; Sun et al., 2019), learning dense representations that capture relational patterns in the graph structure. The field has since progressed to handle complex queries involving multiple entities, relations, and logical operations such as conjunction, disjunction, and existential quantification. Modern *neuro-symbolic* approaches can be broadly categorized based on their query expressiveness: methods supporting smaller subsets of first-order logic (Hamilton et al., 2018; Daza & Cochez, 2020; Ren et al., 2020; Ren & Leskovec, 2020; Arakelyan et al., 2021; 2023; Bai et al., 2023; Zhu et al., 2022) and those handling more general graph pattern queries (Cucumides et al., 2024; Yin et al., 2024).

A fundamental limitation shared by all these approaches is their reliance on *hard logical constraints*: queries can only express conditions that are precisely definable in first-order logic. This restriction prevents users from incorporating nuanced preferences or domain-specific similarity requirements that are naturally expressed through soft constraints. Our work addresses this gap by extending neural query answering to support similarity-based preferences while preserving the logical structure and efficiency of existing methods.

**Similarity constraints for complex queries.** The first work to extend logical query answering with similarity constraints was introduced by Daza et al. (2025), which augmented first-order logic queries with *soft entity constraints* expressing similarity or dissimilarity preferences for query answers. This enriched the expressiveness of methods for CQA by enabling preferences that cannot be captured through logical operators alone. However, we note two key limitations: (i) similarity constraints are only supported on the *target variable* and not on intermediate variables which occur frequently in complex queries, and (ii) similarity incorporation is treated as a *post-processing step* after query evaluation. While post-processing is effective for the target variable, where constraints can be applied directly to the query outputs, it cannot work for intermediate variables that *interact* with query execution. In such cases, uncalibrated updates cannot be meaningfully propagated through neuro-symbolic operators. In contrast, SCORE integrates similarities by mapping them into normalized similarity scores and combining them with fuzzy vector scores, ensuring that updates remain compatible with logical operators.

## 3 PRELIMINARIES

**Knowledge graphs.** We define a knowledge graph as a tuple $\mathcal{G} = (\mathcal{E}, \mathcal{R}, \mathcal{T})$, where $\mathcal{E}$ is the set of entities, $\mathcal{R}$ is the set of binary relations, and $\mathcal{T}$ is the set of triples $(h, r, t)$ with $h, t \in \mathcal{E}$ and $r \in \mathcal{R}$.

**Complex queries.** Queries over KGs are expressed as first-order logic formulas $q(v_1, \dots, v_N)$ with free variables $v_1, \dots, v_N$ ranging over $\mathcal{E}$. Formulas are built from atoms over relations in $\mathcal{R}$ (and their negations), constants denoting specific entities in $\mathcal{E}$, and logical connectives such as conjunction ($\wedge$) and disjunction ($\vee$). For example, consider the question *"What drugs treat diseases that have headache as a symptom?"*. The corresponding query for this question is the following:

$$q(v_1, v_2) = \texttt{SymptomOf}(\texttt{Headache}, v_1) \wedge \texttt{TreatedBy}(v_1, v_2). \tag{1}$$

The free variables in a query can be assigned to specific entities in $\mathcal{E}$. For an assignment $e_1, \dots, e_N \in \mathcal{E}$, we write $q(e_1, \dots, e_N)$ for the resulting **grounded statement**. We write $\mathcal{G} \models q(e_1, \dots, e_N)$ if the set of triples $\mathcal{T}$ in the KG entails this grounded formula.

Continuing the example, suppose $e_1 = \texttt{Migraine}$ and $e_2 = \texttt{Ibuprofen}$. We then obtain the grounded statement:

$$\texttt{SymptomOf}(\texttt{Headache}, \texttt{Migraine}) \wedge \texttt{TreatedBy}(\texttt{Migraine}, \texttt{Ibuprofen}). \tag{2}$$

If both atoms appear in $\mathcal{T}$, then $\mathcal{G} \models q(\texttt{Migraine}, \texttt{Ibuprofen})$.

**Answer set.** Given a query formula $q(v_1, \dots, v_N)$, one may choose a free variable $v_i$ as a designated *target variable*. Denoting as $\boldsymbol{v}_{\neg i}$ all free variables except $v_i$, the answer set $A_i(q)$ is defined as the set of entities that can be assigned to $v_i$ such that the query holds in the graph for some assignment of the remaining variables:

$$A_i(q) = \{e \in \mathcal{E} \mid \exists \boldsymbol{v}_{\neg i} \in \mathcal{E} : \mathcal{G} \models q(\boldsymbol{v}_{\neg i}, v_i = e)\}. \tag{3}$$

Returning to the earlier query

$$q(v_1, v_2) = \texttt{SymptomOf}(\texttt{Headache}, v_1) \wedge \texttt{TreatedBy}(v_1, v_2), \qquad (4)$$

suppose we choose $v_2$ (the drug) as the target variable. Then the answer set is

$$A_2(q) = \{\, e \in \mathcal{E} \mid \exists v_1 \in \mathcal{E} : \mathcal{G} \models \texttt{SymptomOf}(\texttt{Headache}, v_1) \wedge \texttt{TreatedBy}(v_1, e) \,\}. \qquad (5)$$

These definitions form the basis of prior work on complex query answering (CQA), where the goal is to predict answer sets correctly when the graph is incomplete (requiring accurate estimation of the likelihood of entailments $\mathcal{G} \models q(a_1, \ldots, a_N)$) (Hamilton et al., 2018; Daza & Cochez, 2020; Ren et al., 2020; Ren & Leskovec, 2020). They are also adopted by Daza et al. (2025) to introduce similarity constraints over a single target variable.

## 4 SIMILARITY-CONSTRAINED COMPLEX QUERY ANSWERING

We now consider the more general problem where similarity constraints are specified over an arbitrary variable in a query.

**Similarity constraints.** A similarity constraint is intended to capture notions such as "diseases like migraine". Formally, a similarity constraint is a boolean predicate $s$ such that for an entity $e \in \mathcal{E}$, $s(e)$ is true if $e$ meets the similarity constraint, and false otherwise.

**Constrained answer set.** Similarity constraints allow modifying the answer set (Equation (3)) by additionally restricting certain variables to meet the constraint. Given a similarity constraint $s$ applied to variable $v_j$ in a query $q(v_1, \ldots, v_N)$, we denote the *constrained* answer set as follows:

$$\hat{A}_i(q, s, j) = \{ e \in \mathcal{E} \mid \exists \boldsymbol{v}_{\neg i} : (\mathcal{G} \models q(\boldsymbol{v}_{\neg i}, v_i = e)) \wedge s(v_j) \}. \qquad (6)$$

This can be read as the set of entities $e$ that may serve as answers for $v_i$, for which there exists an assignment of the remaining variables such that the query holds in the KG, *and* the assignment to $v_j$ satisfies the similarity constraint $s$.

Because similarity constraints may vary across queries and domain context, $s$ is not fully observed. Instead, we assume access to a small set of $k$ (in the order of tens) labeled examples in a *preference set*:

$$P_s = \{(e_1, s(e_1)), \ldots, (e_k, s(e_k))\}. \qquad (7)$$

Suppose the similarity constraint expresses the notion of *"diseases like migraine"*. Then an example preference set is $P_s = \{(\texttt{Migraine}, \text{true}), (\texttt{ClusterHeadache}, \text{true}), (\texttt{Asthma}, \text{false})\}$.

Together, these definitions provide the foundation for the problem we study in this work.

> **Similarity-constrained Complex Query Answering.** Given a knowledge graph $\mathcal{G}$, a query $q(v_1, \ldots, v_N)$ with target variable $v_i$, and a similarity constraint $s$ applied to variable $v_j$, the task of Similarity-constrained Complex Query Answering (SimCQA) consists of computing the constrained answer set $\hat{A}_i(q, s, j)$ (Equation (6)), where the similarity constraint $s$ is estimated from a preference set $P_s$ (Equation (7)).

This formulation subsumes prior work. If we set $v_i = v_j$, we obtain the setting of Daza et al. (2025) where similarity constraints are applied to a single target variable. If, in addition, we define a similarity function $s(e) = \text{true}$ for all $e \in \mathcal{E}$, we recover standard CQA with purely logical queries.

## 5 SIMILARITY-CONSTRAINED REWEIGHTING

As described in Section 2, several neuro-symbolic methods exist for CQA. A common approach consists of computing one **fuzzy vector** of scores $\mathbf{v}_i \in [0, 1]^{|\mathcal{E}|}$ per variable in a query $q(v_1, \ldots, v_N)$. Each entry in the vector indicates the likelihood of each entity in $\mathcal{E}$ to be assigned to $v_i$. Some examples of methods following this approach are CQD (Arakelyan et al., 2021; 2023), QTO (Bai

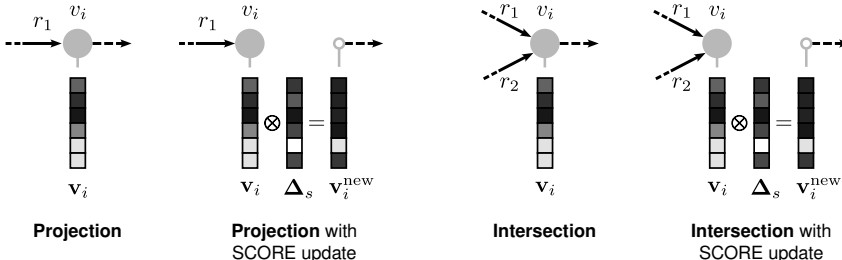

Figure 2: Existing CQA methods compute fuzzy vectors via *projection* and *intersection* operators. We illustrate how we integrate the SCORE update (Equation (10)), by applying it after these operations have been computed by the base CQA method, with $\mathbf{\Delta}_s$ indicating similarity scores.

et al., 2023), GNN-QE (Zhu et al., 2022), and UltraQuery (Galkin et al., 2024). These methods rely on *projection* and *intersection* operators to model the logical query with fuzzy vectors, as illustrated in Figure 2. Once a CQA method has processed a query, the fuzzy vector for the target variable is used as the score for each entity in the KG to answer the query. We describe the fundamental operations used by CQA methods to compute fuzzy vectors for a query in Appendix A.1. Here, we focus on how to incorporate similarity constraints into this computation.

**Similarity-constrained Reweighting (SCORE).** The key difference between CQA and SimCQA is the inclusion of a binary predicate in the constrained answer set that restricts one of the variables $v_j$ to meet the similarity constraint $s$. In principle, this suggests modifying the fuzzy vector $\mathbf{v}_j$ computed by an underlying CQA model by increasing or decreasing the score for each entity depending on whether $s(e)$ is true or false. There are, however, two challenges: (i) the true similarity constraint $s$ is unobserved, and instead it must be approximated from the preference set $P_s$, and (ii) any adjustments made to a fuzzy vector will have an effect in the final answers to the query as the vector is propagated by the underlying CQA method throughout the rest of the query.

We address these challenges with Similarity-constrained Reweighting (SCORE). Intuitively, given a preference set $P_s$ and a fuzzy vector $\mathbf{v}_j$ computed by a base CQA method, SCORE promotes entities that resemble positively labeled, and demotes those resembling negative ones. As we illustrate in Figure 2, we apply updates to a fuzzy vector after the base CQA method has applied a projection or intersection operator. This *reweighted* vector is then propagated through the rest of the query using the base CQA method.

To approximate the true similarity constraint, SCORE relies on entity embeddings obtained from the base CQA method. We employ these embeddings to compute the cosine similarity $\text{sim}(e_1, e_2) \in [-1, 1]$ between pairs of entities $e_1, e_2 \in \mathcal{E}$. In order to combine this value with fuzzy vectors whose elements lie in the interval $[0, 1]$, we compute a *normalized similarity score*, defined as follows:

$$\overline{\text{sim}}(e_1, e_2) = \tfrac{1}{2}\big(\text{sim}(e_1, e_2) + 1\big). \tag{8}$$

This transformation captures the notion that $\overline{\text{sim}}(e_1, e_2) \in [0, 1]$ tends to 0 if the entities are not similar, and to 1 if they are similar.

SCORE computes updates to a fuzzy vector in *logit space*. For a probability $p$, the logit is defined as $\text{logit}(p) = \log(\frac{p}{1-p})$. Logits provide an additive scale for combining independent sources of evidence, which stands in contrast with multiplicative updates that can vanish quickly when values are in the interval $[0, 1]$.

We define $P_s^+$ as the set of entities $e$ in $P_s$ for which $s(e_i) = \text{true}$, and similarly $P_s^-$ for the case where $s(e_i) = \text{false}$. The SCORE update consists of two steps:

1. For each entity $e \in \mathcal{E}$, compute the mean logit similarity:

$$\Delta(P_s^\odot, e) = \frac{1}{|P_s^\odot|} \sum_{e_i \in P_s^\odot} \text{logit}(\overline{\text{sim}}(e_i, e)), \tag{9}$$

   where $P_s^\odot \in \{P_s^+, P_s^-\}$.

2. Reweigh the score of each entity based on the mean logit similarities in logit space, and transform back to the interval $[0, 1]$ with the sigmoid function:

$$\mathbf{v}_j^{\text{new}}[e] = \sigma \left( \text{logit}(\mathbf{v}_j[e]) + w_p \Delta(P_s^+, e) - w_n \Delta(P_s^-, e) \right). \tag{10}$$

SCORE thus only contains two hyperparameters $w_p, w_n \in \mathbb{R}$ which can be tuned to balance the contributions from preferred and non-preferred sets of entities, using a small validation set.

## 5.1 THEORETICAL PROPERTIES

We now highlight three important properties of SCORE. We refer to Appendix A.2 for proofs of the propositions and details on the connection with Bayesian inference.

**Proposition 1** (Monotonicity of SCORE updates). *For any two entities $e_1, e_2 \in \mathcal{E}$, suppose their preference contributions are equal:*

$$w_p \Delta(P_s^+, e_1) - w_n \Delta(P_s^-, e_1) \;=\; w_p \Delta(P_s^+, e_2) - w_n \Delta(P_s^-, e_2).$$

*Then the relative ordering of their fuzzy scores is preserved under the SCORE update:*

$$\mathbf{v}_j[e_1] < \mathbf{v}_j[e_2] \;\;\Longrightarrow\;\; \mathbf{v}_j^{new}[e_1] < \mathbf{v}_j^{new}[e_2].$$

The monotonicity of SCORE contributes to interpretability: whenever the score of entity is promoted or demoted, the exact source of this shift can be traced back to $\Delta(P_s^+, e)$ and $\Delta(P_s^-, e)$. This makes it transparent whether the change is due to evidence from positively or negatively labeled examples, providing a direct explanation of how similarity constraints affect the final scores.

**Proposition 2** (Linear complexity of SCORE). *The computational complexity of applying SCORE is $O(|\mathcal{E}|)$, i.e. linear with respect to the number of entities.*

**Connection with posterior log-odds.** The SCORE update reflects the log-odds update rule in Bayesian inference. This provides a principled justification for our formulation, in which preferences act as additional likelihood terms that adjust the scores of the base CQA model in a coherent way.

## 6 EXPERIMENTS

In our experiments, we follow the ranking-based evaluation which has been employed in works on link prediction (Ji et al., 2022) and complex query answering (Ren et al., 2024). In this setting, the set of triples $\mathcal{T}$ is split into disjoint sets $\mathcal{T}^{\text{train}}$, $\mathcal{T}^{\text{valid}}$, and $\mathcal{T}^{\text{test}}$, and the answer set $A_i(q)$ contains answers that are only reachable by traversing edges in $\mathcal{T}^{\text{test}}$. The goal is to determine the rank assigned by a model to the entities in the answer set $A_i(q)$ when given access to the edges in $\mathcal{T}^{\text{train}} \cup \mathcal{T}^{\text{valid}}$. Similar to prior work (Daza et al., 2025), in SimCQA the goal is the same, with the addition that entities in the *constrained* answer set $\hat{A}_i(q, s, j)$ should be ranked higher than those not in it.

**Datasets.** We conduct experiments on knowledge graphs of different domains and scales: Het-ionet (Himmelstein et al., 2017) is a biomedical KG covering genes, diseases, and drugs, among other biomedical entities. FB15k-237 (Bollacker et al., 2008; Toutanova & Chen, 2015) and NELL995 (Carl-son et al., 2010) are encyclopedic KGs that contain general facts about people, organizations, and locations. We adopt the complex queries of Ren & Leskovec (2020), which cover a wide variety of 14 types of complex queries involving conjunctions, disjunctions, and negations.

We build on the corresponding datasets for SimCQA introduced by Daza et al. (2025), noting that in their setup, entities in preference sets overlap with answers reachable in $\mathcal{T}^{\text{test}}$, turning the problem into a few-shot learning setting. We focus on a more challenging and realistic scenario by restricting preference sets to contain only entities that are only reachable by traversing $\mathcal{T}^{\text{train}} \cup \mathcal{T}^{\text{valid}}$. Furthermore, we extend the datasets with cases of constraints on arbitrary variables of a query, rather than limiting them to the target variable. We present additional details of the datasets in Appendix A.3.

**Baselines.** We run experiments in the settings where similarity constraints are applied to the tar-get variable (**target-variable** SimCQA), and to any variable (**general** SimCQA). We compare the performance of SCORE against NQR (Daza et al., 2025), a neural reranker that, unlike SCORE,

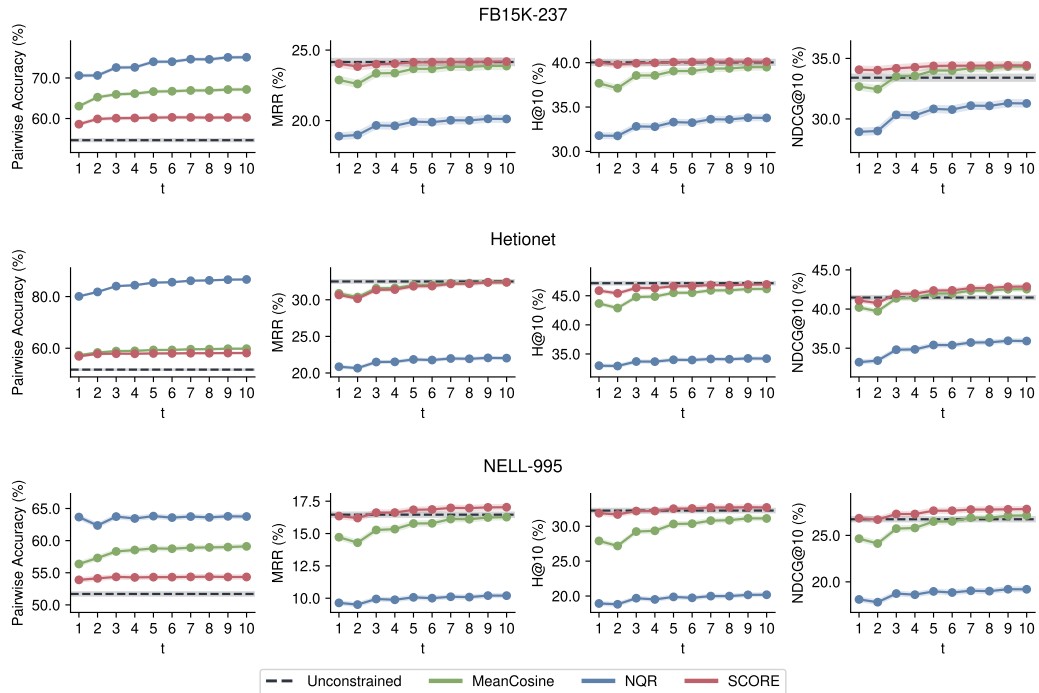

Figure 3: SimCQA results for similarity constraints on the **target** variable. Shaded areas indicate a 95% confidence interval.

requires training with hundreds of thousands of instances of queries and their corresponding constrained answer sets; and MeanCosine: a simple baseline based on raw cosine similarity values, which modifies the fuzzy vector by adding/subtracting the mean cosine similarity with respect to positive/negative examples (Daza et al., 2025). To ensure a fair comparison, we use the same base CQA model (QTO, Bai et al. (2023)) and run an extensive hyperparameter grid search for SCORE, NQR, and MeanCosine separately for each dataset, and select the best values using the validation set. We provide more details in Appendix A.5.

**Metrics.** We evaluate each method in an **interactive setting** (Daza et al., 2025), where preference sets $P_s$ of increasing size $t = 1, \ldots, 10$ are provided and performance is measured at each step. We partition the answer set $A_i(q)$ into answers in the constrained answer set $\hat{A}_i(q, s, j)$, which we denote as $A_i(q)^+$, and those not in it, denoted as $A_i(q)^-$. We then report:

- **MRR** and **Hits@k (H@k)** measure whether a method preserves the ranking quality of the base CQA model, i.e., how well entities in the unconstrained answer set $A_i(q)$ are ranked.

- **Pairwise Accuracy (PA)** isolates similarity-constraint satisfaction, checking only whether entities in the constrained set $\hat{A}_i(q, s, j)$ are ranked above those in $A_i(q)^-$.

- **NDCG@k** combines both aspects by assigning graded relevance (0 to non-answers, 1 to $A_i(q)^-$, and 2 to $A_i(q)^+$), rewarding rankings that balance global quality with constraint satisfaction.

Formal definitions and additional discussion of these metrics are provided in Appendix A.4.

### 6.1 RESULTS

**Target-variable SimCQA.** We present results for target-variable SimCQA in Figure 3 averaged across all queries and indicating a 95% confidence interval. We report the metrics of the base CQA method (QTO) as a reference without similarity constraints (shown as Unconstrained).

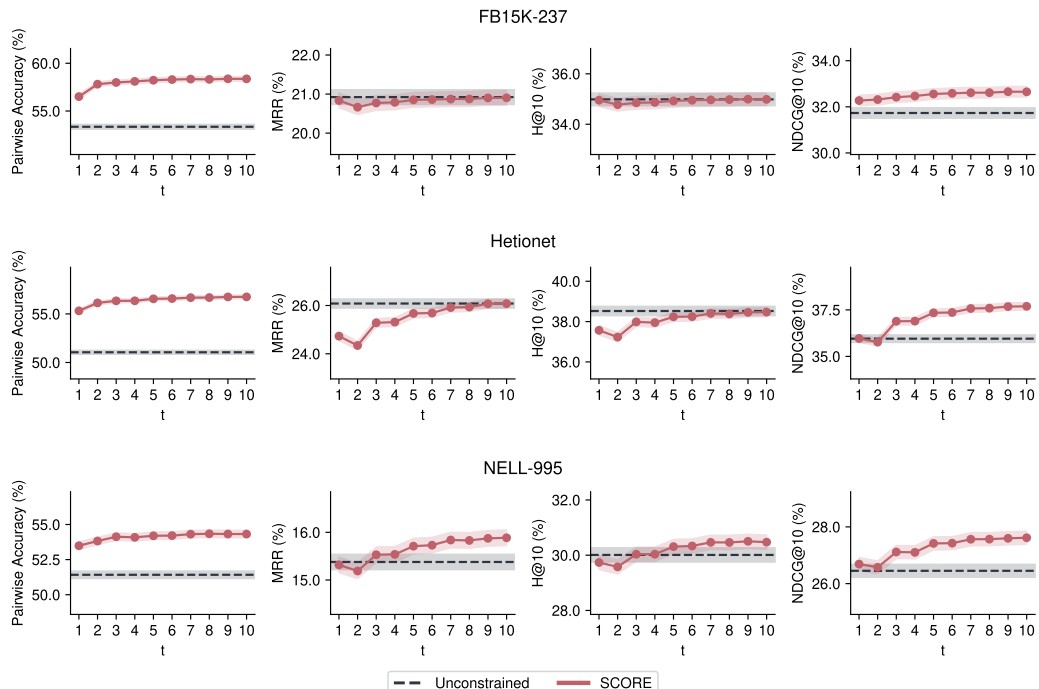

Figure 4: SimCQA results for similarity constraints on the **target and intermediate** variables. Shaded areas indicate a 95% confidence interval.

SCORE achieves the highest NDCG@10 across all datasets, indicating that its resulting ranking more closely resembles the optimal ranking where preferred answers are ranked above non-preferred ones, which are in turn ranked higher than entities not answering a query. SCORE performs particularly well on small preference sets, indicating that it can quickly adapt to a few labeled examples.

The MeanCosine baseline shows a competitive NDCG@10, though the gap with SCORE increases for small preference sets. The higher pairwise accuracy and lower MRR and H@10 show that a significant contributor to the NDCG@10 stems from a good separation of preferred vs. non-preferred entities, at a cost of lower global ranking quality.

The neural NQR model performs better at capturing preferences, but it does so in a more aggressive way that harms global ranking quality. While NQR is designed to balance the two goals (Daza et al., 2025), our results show that the simpler SCORE update works better than the potentially non-linear effects introduced by NQR. Since we run experiments on a more challenging benchmark aimed at a better separation between training and test instances, the difference may also indicate that the generalization properties of NQR are limited.

**General SimCQA.** We present results for SimCQA over arbitrary variables in a query in Figure 4. Since this case requires normalized score updates that can be combined with the base CQA model for further propagation in query execution, SCORE is the only model applicable in this setting. We note that the performance of SCORE is consistent, though slightly lower, than in the case of similarity constraints on the target variable. The steady increase of NDCG@10 indicates that as the preference set becomes larger, preferred entities are ranked higher while preserving global ranking quality.

**Per-query type results.** We present more detailed results of SimCQA performance for each of the 14 query types we consider in Tables 5 and 6 in Appendix A.6. Both in target-variable as well as general SimCQA, we observe that for the large majority of query structures, SCORE results in the best performance. We identify three query types involving intersections where MeanCosine results in better performance by a small margin, but this does not point to a general trend across datasets.

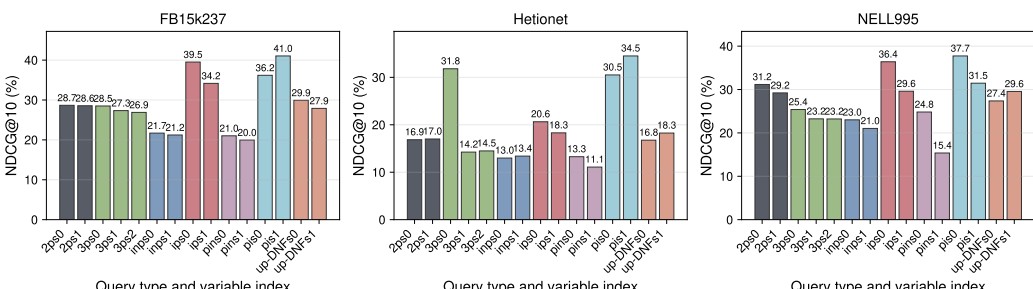

Figure 5: NDCG@10 per query type and variable position on different datasets. Indices indicate where the positions of variables in which similarity constraints are applied in the query.

Table 1: Example 2-hop query from Hetionet, showing the top-scored entities for both the intermediate (drug) and target (side effect) variables before and after applying similarity constraints with SCORE. ▲ indicates a promoted entity, ▼ a demoted one.

| $q(v_1, v_2) = \texttt{TreatedBy}(\text{Migraine}, v_1) \wedge \texttt{SideEffect}(v_1, v_2)$ | | | |
|---|---|---|---|
| $P_s^+ = \{\text{Diclofenac, Naproxen, Prednisone}\}, P_s^- = \{\text{Rizatriptan, Orphenadrine, Ergotamine}\}.$ | | | |
| **Initial Top-5** | | **Top-5 after SCORE** | |
| $v_1$ | $v_2$ | $v_1$ | $v_2$ |
| Sumatriptan | Hemiplegia transient | ▲ Diclofenac | ▲ Hypersensitivity |
| Ergotamine | Cluster headache | ▲ Naproxen | ▲ Dermatitis |
| Frovatriptan | Arteriospasm coronary | ▲ Prednisone | ▲ Pain |
| Antipyrine | Temporal arteritis | ▼ Sumatriptan | ▲ Headache |
| Naratriptan | Hypertensive episode | ▼ Frovatriptan | ▲ Asthma |
| | NDCG@10: 28.4 | | NDCG@10: 36.5 |

**Influence of variable position.** Figure 5 reports NDCG@10 results of SCORE when broken down by types of queries containing more than one variable, and by the variable position on which similarity constraints are applied. We specify an index that indicates the topological order of variables in the query (see Figure 6 in Appendix A.3). For example, in a 2-hop query, 2ps0 applies similarity constraints to the intermediate variable, while 2ps1 applies them to the target variable. Interestingly, we observe that in most cases constraints applied *earlier* in the query (i.e., on intermediate variables) yield slightly higher performance than when applied to later variables or directly to the target variable. This suggests that preference information is more effectively propagated when introduced at earlier stages of the query, reinforcing the value of supporting constraints on arbitrary variables beyond the target.

**Runtime overhead.** Table 2 reports average query execution times (in ms) with and without SCORE. Across datasets, the additional cost is modest, ranging from 1.24 up to 8.45 ms per query. Importantly, the overhead does not grow superlinearly with dataset size, confirming our analysis that SCORE is linear in the number of entities. SCORE can thus be deployed in practice without compromising the efficiency of existing CQA methods.

Table 2: Average runtime per query (ms) with and without SCORE.

| Dataset | Base | +SCORE | $\Delta$ |
|---|---|---|---|
| FB15k237 | 2.96 | 4.20 | +1.24 |
| Hetionet | 25.65 | 28.68 | +3.03 |
| NELL-995 | 75.51 | 83.96 | +8.45 |

**Qualitative example.** Table 1 illustrates how SCORE applies similarity constraints to an intermediate variable on a query in the Hetionet, corresponding to the question *"What side effects are associated with drugs that treat migraine?"*. The base model primarily retrieves drugs from the classes triptans and ergots, which are used for advanced treatment of migraine, together with unrelated side effects achieving an NDCG@10 of 28.4. After providing a preference set favoring anti-inflammatory drugs and steroids (used in initial stages), SCORE shifts the ranking to compounds of these classes and their

associated side effects, improving NDCG@10 to 36.5. This example highlights the interpretability of SCORE and its ability to propagate similarity evidence to improve the quality of the answers.

## 7 DISCUSSION

Our results demonstrate that SCORE provides an effective and broadly applicable mechanism for incorporating similarity constraints into complex query answering. We observe that preference information can be propagated through the query without harming global ranking quality. With these results, SCORE has a broad scope of implications: it extends the expressivity of CQA systems beyond purely logical constraints, it can be easily adapted to existing CQA methods by requiring tuning only two lightweight parameters, and it provides interpretable score updates.

While we focused on specifying preferences for one variable at a time, a natural extension is to allow similarity constraints on multiple variables simultaneously. This substantially increases the number of possible configurations: for 2p queries one may constrain the intermediate variable, the target variable, or both; for 3p queries this grows to seven combinations; and in general, for a query with $N$ variables, the number of variable combinations is $2^N - 1$. Systematically evaluating these combinations is an interesting direction for future work.

The benchmarks introduced by Daza et al. (2025) derive preference sets using textual semantic similarity, but in some domains numerical and other types of attributes may provide an alternative signal. While textual descriptions often encode high-level semantics, future work could explore similarity constraints grounded in other data types.

Finally, SCORE extends a broad class of neuro-symbolic CQA systems that traverse the query graph and compute fuzzy score vectors for each variable, including QTO (Bai et al., 2023), CQD (Arakelyan et al., 2021), GNN-QE (Zhu et al., 2022), and UltraQuery (Galkin et al., 2024), among others. These methods expose intermediate scores for each variable, which allows similarity constraints to be injected at specific points in the query. In contrast, embedding-based methods embed the entire query into a single vector (Hamilton et al., 2018; Daza & Cochez, 2020; Zhang et al., 2021; 2024), making it non-trivial to incorporate similarity constraints on arbitrary variables. Investigating how to adapt such methods is an interesting direction for future work.

## 8 CONCLUSION

We introduce and formalize the general problem of Similarity-constrained Complex Query Answering (SimCQA), extending the expressivity of logical queries by allowing similarity constraints on arbitrary variables. To address this problem, we propose Similarity-Constrained Reweighting (SCORE), a computationally efficient and interpretable method that integrates similarity constraints consistently with the computations of a base method for complex query answering. Experiments across multiple knowledge graphs show that SCORE is capable of capturing similarity constraints without harming– and sometimes even improving–the answers to a query. We further find that constraints applied earlier in the query often yield stronger gains by allowing preference information to propagate more effectively. Taken together, these results highlight SCORE as a practical and general approach for more expressive methods for CQA supporting mechanisms for specifying constraints beyond first order logic. For future work, we envision extending expressivity by incorporating similarities derived from natural language, opening the door to richer forms of guidance and more natural interfaces to knowledge graphs.

# 9 Reproducibility Statement

We release an anonymous code repository with scripts to run SCORE and all baselines, along with dataset splits and configuration files for every experiment (Section 6; link in the abstract). The operators and base CQA pipeline needed to reproduce our method are specified in Appendix A.1. Further details on the SimCQA benchmark can be found in Appendix A.3, with dataset statistics in Tables 3 and 4. All model and search spaces used for SCORE, NQR, and MeanCosine are enumerated in Appendix A.5; we also provide the exact hyperparameters selected per dataset in job files in the repository. The theoretical properties of SCORE (monotonicity, linear-time overhead, and the log-odds connection) are stated in Section 5.1 with complete proofs in Appendix A.2. Figures 3 and 4 report means with 95% confidence invertals; the code includes the evaluation and plotting scripts that regenerate these figures from raw runs. Finally, we include a README to facilitate replication of our experimental results.

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

# A APPENDIX

## A.1 NEURO-SYMBOLIC METHODS FOR CQA

This section provides an overview of the logical operators in the logical queries we consider in this work, as well as the relaxed operators used to model logical operators in existing methods for neuro-symbolic CQA that can be used as base methods for SimCQA.

**Logical operators.** We consider logical queries over a KG $\mathcal{G} = (\mathcal{E}, \mathcal{R}, \mathcal{T})$ specified by a formula $q(v_1, \ldots, v_N)$ in first-order logic, written in disjunctive normal form (DNF), i.e., as a disjunction ($\vee$) of conjunctions ($\wedge$) of *atoms*:

$$q(v_1, \ldots, v_N) = (c_1^1 \wedge \cdots \wedge c_{m_1}^1) \vee \cdots \vee (c_1^n \wedge \cdots \wedge c_{m_n}^n), \tag{11}$$

where each $c_j^i$ denotes a predicate applied to two variables, or a variable and a known entity in $\mathcal{E}$, while optionally including negations:

$$c_j^i = \begin{cases} r(e, v) \text{ or } r(v', v) \\ \neg r(e, v) \text{ or } \neg r(v', v) \end{cases} \quad \text{with } e \in \mathcal{E}, v, v' \in \{v_1, \ldots, v_N\}, \text{ and } r \in \mathcal{R}. \tag{12}$$

**T-norms.** Neuro-symbolic methods for CQA incorporate logical operators in their computations by relaxing them into continuous functions known as **t-norms** and **t-conorms** from fuzzy logics (Zadeh, 1965), where values in $[0, 1]$ model degrees of truth. In this setting, conjunction ($\wedge$) is generalized by a *t-norm* $\top : [0, 1]^2 \to [0, 1]$, disjunction ($\vee$) by a *t-conorm* $\bot : [0, 1]^2 \to [0, 1]$, and negation ($\neg$) by the standard fuzzy complement $1 - x$. Using these relaxations, a query $q(v_1, \ldots, v_N)$ in disjunctive normal form can be expressed as

$$q(v_1, \ldots, v_N) = (c_1^1 \top \ldots \top c_{m_1}^1) \bot \ldots \bot (c_1^n \top \ldots \top c_{m_n}^n), \tag{13}$$

where each atom $c_j^i$ is mapped to a fuzzy truth value in $[0, 1]$.

An example is the *product* t-norm and its dual t-conorm, defined as

$$\top(x, y) = x \cdot y, \tag{14}$$
$$\bot(x, y) = x + y - x \cdot y, \tag{15}$$

which recover classical Boolean logic when $x, y \in \{0, 1\}$. Several other t-norms exist, for an in-depth description see Hájek (1998).

**Fuzzy vectors for CQA.** Several neuro-symbolic methods assign a vector $\mathbf{v}_i \in [0, 1]^{|\mathcal{E}|}$ to each variable $v_i$ in a query, which represents the likelihood that the variable is assigned each entity in $\mathcal{E}$ (Arakelyan et al., 2021; 2023; Bai et al., 2023; Zhu et al., 2022; Cucumides et al., 2024; Yin et al., 2024). Similarly, constants in a query can be represented as one hot vectors $\mathbf{c} \in \{0, 1\}^{|\mathcal{E}|}$, such that $\mathbf{c}[e] = 1$ for a constant entity $e$, and all other entries are zero. This vector can then be transformed with t-norms, t-conorms, and negations depending on the query being answered.

**Projections and negations.** For a query formula $q(v_1, v_2) = r(v_1, v_2)$ with $r \in \mathcal{R}$, a fuzzy vector $\mathbf{v}_1$ is projected via a function $f_r(\mathbf{v}_1) = \mathbf{v}_2 \in [0, 1]^{|\mathcal{E}|}$ which yields the fuzzy vector for $v_2$. Often the function $f_r$ is a parameterized neural network optimized for 1-hop link prediction (Arakelyan et al., 2021; 2023; Bai et al., 2023) or CQA (Zhu et al., 2022; Galkin et al., 2024). For a negation $q(v_1, v_2) = \neg r(v_1, v_2)$, the fuzzy vector is computed as $\mathbf{1} - f_r(\mathbf{v}_1)$, where $\mathbf{1}$ is an all-ones vector of length $|\mathcal{E}|$.

**Conjunctions and disjunctions.** For a conjunction $q(v_1, v_2, v_3) = r_1(v_1, v_2) \wedge r_2(v_3, v_2)$ with $r_1, r_2 \in \mathcal{R}$ the two predicates are first projected as $f_{r1}(\mathbf{v}_1)$ and $f_{r2}(\mathbf{v}_3)$, and the fuzzy vector of $\mathbf{v}_2$ is computed via the t-norm: $\mathbf{v}_2 = f_{r1}(\mathbf{v}_1) \top f_{r1}(\mathbf{v}_3)$. For a disjunction $r_1(v_1, v_2) \wedge r_2(v_3, v_2)$ the result is computed via the t-conorm, $\mathbf{v}_2 = f_{r1}(\mathbf{v}_1) \bot f_{r1}(\mathbf{v}_3)$.

## A.2 PROOFS OF PROPERTIES OF SCORE

**Proposition 1** (Monotonicity of SCORE updates). *For any two entities $e_1, e_2 \in \mathcal{E}$, suppose their preference contributions are equal:*

$$w_p \Delta(P_s^+, e_1) - w_n \Delta(P_s^-, e_1) \;=\; w_p \Delta(P_s^+, e_2) - w_n \Delta(P_s^-, e_2).$$

*Then the relative ordering of their fuzzy scores is preserved under the SCORE update:*

$$\mathbf{v}_j[e_1] < \mathbf{v}_j[e_2] \;\implies\; \mathbf{v}_j^{new}[e_1] < \mathbf{v}_j^{new}[e_2].$$

*Proof.* By definition of the update in Equation (10),

$$\mathbf{v}_j^{new}[e] = \sigma(\mathrm{logit}(\mathbf{v}_j[e]) + \delta(e)),$$

where $\delta(e) = w_p \Delta(P_s^+, e) - w_n \Delta(P_s^-, e)$. If $\delta(e_1) = \delta(e_2)$, then the same shift is applied to both $\mathrm{logit}(\mathbf{v}_j[e_1])$ and $\mathrm{logit}(\mathbf{v}_j[e_2])$. Since $\sigma(\cdot)$ is strictly monotone, the ordering of the original logits is preserved, and so is the ordering of the values in the fuzzy vector. Hence SCORE is monotonic with respect to the scores of the underlying model: it preserves the base model's ranking whenever preference contributions are equal. □

The monotonicity of SCORE contributes to interpretability: whenever the score of entity is promoted or demoted, the exact source of this shift can be traced back to $\Delta(P_s^+, e)$ and $\Delta(P_s^-, e)$. This makes it transparent whether the change is due to evidence from positively or negatively labeled examples, providing a direct explanation of how similarity constraints affect the final scores.

**Proposition 2** (Linear complexity of SCORE). *The computational complexity of applying SCORE is $O(|\mathcal{E}|)$, i.e. linear with respect to the number of entities.*

*Proof.* Let $|\mathcal{E}|$ be the number of entities and $k$ the number of examples in the preference set. For each candidate entity $e \in \mathcal{E}$, SCORE computes normalized similarities $\overline{\mathrm{sim}}(e, e_i)$ with all $k$ labeled examples, which costs $O(kd)$ if embeddings are $d$-dimensional. Aggregating these values and computing the update in Equation (10) also requires $O(k)$. Thus the cost per entity is $O(kd)$, and the total cost is $O(|\mathcal{E}|kd)$. Since $k$ is small (on the order of tens of examples) and $d$ is fixed by the underlying embedding model, both factors are constant in practice. Therefore the overall complexity scales linearly with $|\mathcal{E}|$. □

**Connection with posterior log-odds.** Assuming a probabilistic interpretation of scores, we reinterpret the similarity constraint $s(e)$ (previously a binary predicate) as the probability that an entity $e \in \mathcal{E}$ is to satisfy the constraint. For brevity, let $\mathbf{v} = \mathbf{v}_j[e]$ denote the base CQA score for entity $e$. The SCORE update can then be written as

$$\mathbf{v}^{new} = \sigma(\mathrm{logit}(\mathbf{v}) + \mathrm{logit}(s(e))). \tag{16}$$

Taking logits gives

$$\mathrm{logit}(\mathbf{v}^{new}) = \mathrm{logit}(\mathbf{v}) + \mathrm{logit}(s(e)), \tag{17}$$

and by the definition of the logit,

$$\log \frac{\mathbf{v}^{new}}{1 - \mathbf{v}^{new}} = \log \frac{\mathbf{v}}{1 - \mathbf{v}} + \log \frac{s(e)}{1 - s(e)}. \tag{18}$$

This is equivalent to the additive update of log-odds in Bayesian inference: the posterior log-odds equal the prior log-odds plus the log-likelihood ratio of the evidence. More concretely, let $x$ be the random variable indicating whether $e$ is an answer. If we view $\mathbf{v} = P(x = 1)$ as the prior from the CQA model and $s(e)$ as a data-dependent score that encodes the strength of evidence for $x = 1$ versus $x = 0$, then

$$\log \frac{P(x = 1 \mid \mathrm{data})}{P(x = 0 \mid \mathrm{data})} = \log \frac{P(x = 1)}{P(x = 0)} + \log \frac{P(\mathrm{data} \mid x = 1)}{P(\mathrm{data} \mid x = 0)}. \tag{19}$$

Thus, the SCORE update mirrors the Bayesian log-posterior update when scores are interpreted probabilistically. While the base CQA model and similarity function may not produce calibrated probabilities, operating in logit space ensures that preference evidence is combined additively and propagated consistently with the logical operators of the query.

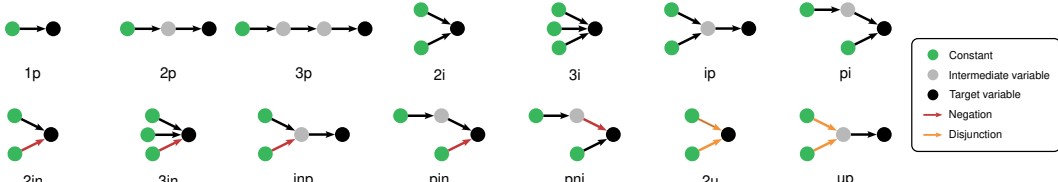

Figure 6: Query graphs of different types covered in the datasets used in our experiments, along with their designated name.

### A.3 DATASETS

In this section we provide additional details on how we construct a generalized benchmark for SimCQA with similarity constraints on arbitrary variables.

**Base resources.** We start from standard benchmarks for complex query answering (Ren & Leskovec, 2020), which cover 14 query types involving conjunction, disjunction, and negation (illustrated in Figure 6). For each knowledge graph (Hetionet, FB15k-237, and NELL995), we generate training, validation, and test queries following the same procedure as in prior work. Each query is associated with a designated *target variable* $v_i$ whose answers form the set $A_i(q)$.

**Preference sets and constrained answers.** Following Daza et al. (2025), we rely on clusterings of embeddings as a proxy for unobserved similarity constraints. For each query, we (i) identify a variable $v_j$ (which may be the target or any intermediate variable), (ii) collect the entities that appear as valid bindings for $v_j$, and (iii) apply hierarchical agglomerative clustering to obtain two partitions of this set. These partitions define preference sets of the form $P_s = \{(e, \texttt{true}) : e \in C^+\} \cup \{(e, \texttt{false}) : e \in C^-\}$, where $C^+$ and $C^-$ are clusters. The induced constrained answer set $\hat{A}_i(q, s, j)$ is then computed by propagating the partition of $v_j$ through the query, yielding a corresponding partition of the target answers.

While clusters derived from embeddings are not the only or definitive notion of similarity, they offer a systematic, reproducible, and scalable way to approximate semantic groupings, allowing us to study preference-based reranking behavior across a large collection of queries. Their use as a proxy is supported by prior work showing that distances in embedding spaces correlate with human perceptions of semantic similarity and relatedness (Reimers & Gurevych, 2019; Grand et al., 2022; Merkx et al., 2022), which we observed also occur in the datasets in our experiments.

**Generalization protocol.** A key difference with Daza et al. (2025) is how we split answers across $\mathcal{T}^{\text{train}}$, $\mathcal{T}^{\text{valid}}$, and $\mathcal{T}^{\text{test}}$. In their original setup, preference examples could overlap with answers reachable in $\mathcal{T}^{\text{test}}$, meaning that at test time the model effectively observes part of the ground truth answers (a few-shot learning scenario). To ensure stricter generalization, we impose the following rules:

- The ground truth constrained answer set $\hat{A}_i(q, s, j)$ is defined only with respect to $\mathcal{T}^{\text{test}}$. This ensures that evaluation reflects the model's ability to generalize to unseen facts.
- Preference sets $P_s$ are restricted to contain only entities reachable via $\mathcal{T}^{\text{train}} \cup \mathcal{T}^{\text{valid}}$. This prevents test leakage and forces the model to extrapolate similarity constraints to novel cases at test time.

**Constraints on arbitrary variables.** Whereas Daza et al. (2025) considered constraints only on the target variable, we generalize the construction to allow constraints on *any* variable in the query. This is achieved by clustering the bindings of each free variable $v_j$ in the query (not just the target), and retaining those partitions that induce non-trivial and non-overlapping partitions of the target answer set. This extension produces preference/evidence pairs where feedback is given on intermediate variables, leading to richer scenarios for similarity-constrained reasoning.

**Filtering and quality control.** To ensure meaningful preference sets, we apply the following filters: (i) only variables with at least 10 and at most 100 distinct bindings are considered for clustering; (ii)

Table 3: Statistics of the datasets used in our experiments.

| Dataset | Knowledge Graph | | | Queries | | | Preference Sets | | |
|---|---|---|---|---|---|---|---|---|---|
| | Entities | Relations | Edges | Train | Validation | Test | Train | Validation | Test |
| FB15k237 | 14,505 | 237 | 310,079 | 157,479 | 20,961 | 21,352 | 580,623 | 68,834 | 70,365 |
| Hetionet | 45,158 | 24 | 2,250,198 | 91,820 | 29,475 | 29,488 | 325,443 | 99,785 | 99,284 |
| NELL995 | 63,361 | 200 | 142,804 | 75,446 | 17,289 | 17,435 | 245,511 | 52,446 | 52,589 |

Table 4: Statistics of the queries and preference sets in the datasets used in our experiments.

| Structure | 1p | 2p | 3p | 2i | 3i | ip | pi | 2in | 3in | inp | pin | pni | 2u | up | Total |
|---|---|---|---|---|---|---|---|---|---|---|---|---|---|---|---|
| **FB15k237** | | | | | | | | | | | | | | | |
| *Training* | | | | | | | | | | | | | | | |
| Queries | 7,596 | 34,720 | 56,495 | 16,205 | 8,870 | 0 | 0 | 5,300 | 3,821 | 9,384 | 8,891 | 6,197 | 0 | 0 | 157,479 |
| Pref. Sets | 19,075 | 121,420 | 240,770 | 46,302 | 26,265 | 0 | 0 | 16,536 | 11,530 | 41,128 | 38,217 | 19,380 | 0 | 0 | 580,623 |
| *Validation* | | | | | | | | | | | | | | | |
| Queries | 1,631 | 1,746 | 2,255 | 1,341 | 753 | 1,249 | 1,501 | 1,210 | 1,560 | 2,184 | 1,825 | 782 | 1,087 | 1,837 | 20,961 |
| Pref. Sets | 4,149 | 5,967 | 8,876 | 3,972 | 2,275 | 4,099 | 5,239 | 3,516 | 4,549 | 8,140 | 6,256 | 2,182 | 3,251 | 6,363 | 68,834 |
| *Test* | | | | | | | | | | | | | | | |
| Queries | 1,934 | 1,776 | 2,267 | 1,347 | 814 | 1,337 | 1,399 | 1,191 | 1,553 | 2,133 | 1,844 | 814 | 1,080 | 1,863 | 21,352 |
| Pref. Sets | 4,898 | 6,157 | 9,138 | 3,875 | 2,398 | 4,358 | 4,834 | 3,456 | 4,581 | 8,017 | 6,516 | 2,356 | 3,206 | 6,575 | 70,365 |
| **Hetionet** | | | | | | | | | | | | | | | |
| *Training* | | | | | | | | | | | | | | | |
| Queries | 19,595 | 20,075 | 2,000 | 20,075 | 20,075 | 0 | 0 | 2,000 | 2,000 | 2,000 | 2,000 | 2,000 | 0 | 0 | 91,820 |
| Pref. Sets | 66,076 | 79,488 | 9,127 | 68,553 | 66,012 | 0 | 0 | 7,281 | 6,768 | 7,503 | 7,586 | 7,049 | 0 | 0 | 325,443 |
| *Validation* | | | | | | | | | | | | | | | |
| Queries | 9,975 | 1,500 | 1,500 | 1,500 | 1,500 | 1,500 | 1,500 | 1,500 | 1,500 | 1,500 | 1,500 | 1,500 | 1,500 | 1,500 | 29,475 |
| Pref. Sets | 32,653 | 5,185 | 5,805 | 5,024 | 5,017 | 5,040 | 5,801 | 5,158 | 5,068 | 5,223 | 4,970 | 4,829 | 4,877 | 5,135 | 99,785 |
| *Test* | | | | | | | | | | | | | | | |
| Queries | 9,988 | 1,500 | 1,500 | 1,500 | 1,500 | 1,500 | 1,500 | 1,500 | 1,500 | 1,500 | 1,500 | 1,500 | 1,500 | 1,500 | 29,488 |
| Pref. Sets | 32,865 | 5,049 | 5,796 | 5,056 | 4,980 | 4,843 | 5,783 | 5,092 | 5,121 | 5,218 | 4,789 | 4,630 | 4,820 | 5,242 | 99,284 |
| **NELL995** | | | | | | | | | | | | | | | |
| *Training* | | | | | | | | | | | | | | | |
| Queries | 1,871 | 17,649 | 24,859 | 4,547 | 3,034 | 0 | 0 | 3,848 | 2,459 | 5,824 | 6,016 | 5,339 | 0 | 0 | 75,446 |
| Pref. Sets | 4,677 | 50,328 | 92,914 | 11,221 | 7,530 | 0 | 0 | 11,507 | 7,151 | 22,040 | 22,145 | 15,998 | 0 | 0 | 245,511 |
| *Validation* | | | | | | | | | | | | | | | |
| Queries | 942 | 1,281 | 1,532 | 824 | 468 | 836 | 1,042 | 1,381 | 1,383 | 2,126 | 1,859 | 1,058 | 1,152 | 1,405 | 17,289 |
| Pref. Sets | 2,254 | 4,043 | 5,159 | 2,292 | 1,291 | 2,410 | 2,855 | 3,945 | 4,214 | 7,229 | 6,038 | 3,046 | 3,270 | 4,400 | 52,446 |
| *Test* | | | | | | | | | | | | | | | |
| Queries | 1,004 | 1,253 | 1,607 | 993 | 635 | 772 | 1,039 | 1,344 | 1,510 | 2,106 | 1,770 | 938 | 1,106 | 1,358 | 17,435 |
| Pref. Sets | 2,437 | 3,917 | 5,514 | 2,659 | 1,789 | 2,077 | 2,933 | 3,860 | 4,479 | 7,182 | 5,743 | 2,618 | 3,076 | 4,305 | 52,589 |

both positive and negative clusters must induce at least 5 valid answers on the training edges; (iii) induced answer sets must be non-empty and disjoint. These criteria discard degenerate cases and guarantee that similarity constraints provide useful supervisory signals.

**Output.** Each dataset instance consists of:

1. a query $q$ and target variable $v_i$,

2. the variable $v_j$ on which preferences are applied,

3. a preference set $P_s$ built from clusters of bindings of $v_j$,

4. the ground truth constrained answer set for the target variable $v_i$: $\hat{A}_i(q, s, j)$ derived from $\mathcal{T}^{\text{test}}$.

We repeat this process across all query types and variables, producing a diverse benchmark that covers both biomedical and encyclopedic domains. Dataset statistics are reported in Table 3 and Table 4.

## A.4 METRICS

**Global ranking quality.** For each entity in the unconstrained answer set $A_i(q)$, we determine its rank $r$, and compute the Mean Reciprocal Rank (MRR) and Hits@k (H@k), computed per answer as $\text{MRR} = \frac{1}{r}$ and $\text{H@k} = \mathbb{1}[r \leq k]$ (where $\mathbb{1}$ is an indicator function) and averaged over all queries.

**Similarity constraint satisfaction.** Here we measure whether methods rank preferred answers higher than non-preferred ones. We partition the answer set $A_i(q)$ into answers in the constrained answer set $\hat{A}_i(q, s, j)$, which we denote as $A_i(q)^+$, and those not in it, denoted as $A_i(q)^-$. We then compute Pairwise Accuracy:

$$\text{PA} = \sum_{e^+ \in A_i(q)^+} \sum_{e^- \in A_i(q)^-} \mathbb{1}\left[r(e^+) < r(e^-)\right], \tag{20}$$

where $r(e)$ indicates the ranking of entity $e$.

We also compute the Normalized Discounted Cumulative Gain at $k$ (NDCG@k), assigning 0 relevance to non-answers, 1 to answers in $A_i(q)^-$, and 2 to answers in $A_i(q)^+$. Concretely, the discounted cumulative gain is

$$\text{DCG@}k = \sum_{e \in A_i(q)} \frac{2^{\text{rel}(e)} - 1}{\log_2(j + 1)}, \tag{21}$$

where

$$\text{rel}(e) = \begin{cases} 0 & \text{if } e \notin A_i(q) \\ 1 & \text{if } \in A_i(q) \setminus \hat{A}_i(q, s, j) \\ 2 & \text{if } \in \hat{A}_i(q, s, j) \end{cases} \tag{22}$$

and the normalized score is

$$\text{NDCG@}k = \frac{\text{DCG@}k}{\text{IDCG@}k}, \tag{23}$$

where IDCG@k is the maximum achievable DCG under the given relevance scores. Since NDCG@k is normalized by the ideal ranking, a value of 1 indicates that all preferred answers appear above non-preferred answers, which in turn appear above non-answers. We report the average of PA and NDCG@k over all queries.

## A.5 Experimental details

**SCORE.** The only two parameters that need to be tune in SCORE are the weights $w_p$ and $w_n$ in Equation (10) that control the strength of shifts in logit space to the base score due to similarities with the preference set. We run a grid search with values in $\{0.25, 0.5, 1.0\}$, for a total of 9 possible combinations of $w_p$ and $w_n$.

**NQR.** We perform a grid search with values of the learning rate in $\{1 \times 10^{-5}, 1 \times 10^{-4}\}$, the margin in the preference loss in $\{0.05, 0.1, 0.25\}$, and the KL divergence weight in $\{0.1, 1.0, 10\}$, resulting in a total of 18 possible combinations of hyperparameters. As done by Daza et al. (2025), we train NQR on queries of type 1p, as training on more query types is more expensive but brings little benefits.

**MeanCosine.** The update rule in the MeanCosine baseline is given by the following expression:

$$\mathbf{v}_i^{\text{new}}[e] = \alpha \mathbf{v}_i[e] + (1 - \alpha) \left( \frac{1 + \beta}{2|P_s^+|} \sum_{e_i \in P_s^+} \text{sim}(e_i, e) - \frac{1 - \beta}{2|P_s^-|} \sum_{e_i \in P_s^-} \text{sim}(e_i, e) \right). \tag{24}$$

Intuitively, MeanCosine computes a convex combination of the original score, and a score due to the mean of raw cosine similarity values, which is balanced with the hyperparameter $\alpha \in [0, 1]$. Similarities from $P_s^+$ are added, and those from $P_s^-$ are subtracted, and the balance of these two is determined by the value of $\beta \in [-1, 1]$. We run a grid search for the values of $\alpha$ in $\{0.25, 0.5, 0.75\}$, and $\beta \in \{-0.5, 0, 0.5\}$ for a total of 9 possible configurations.

## A.6 Additional results

We present more detailed results on SimCQA performance for each of the 14 types of complex queries we consider in our work (shown in Figure 6).

| Method | 1p | 2in | 2i | 2p | 2u | 3in | 3i | 3p | inp | ip | pin | pi | pni | up | Avg. |
|---|---|---|---|---|---|---|---|---|---|---|---|---|---|---|---|
| | | | | | | | FB15k237 | | | | | | | | |
| Unconstrained | 43.90 | 23.34 | 47.40 | 28.75 | 34.37 | 37.14 | 50.54 | 27.16 | 21.48 | 34.32 | 20.05 | 41.32 | 9.87 | 27.98 | 31.97 |
| MeanCosine | 44.25 | 23.76 | 47.92 | 29.67 | 34.80 | 37.20 | 50.92 | 26.95 | **22.81** | 34.29 | 20.38 | 40.53 | 10.33 | 28.34 | 32.30 |
| NQR | 41.61 | 21.13 | 46.13 | 25.24 | 32.33 | 33.82 | 49.43 | 21.30 | 19.48 | 27.79 | 16.92 | 36.12 | 9.25 | 25.95 | 29.04 |
| SCORE | **44.43** | **24.07** | **48.06** | **29.84** | **34.91** | **37.80** | **51.38** | **28.34** | 22.62 | **35.61** | **21.20** | **42.03** | **10.57** | **28.73** | **32.83** |
| | | | | | | | Hetionet | | | | | | | | |
| Unconstrained | 52.95 | 33.11 | 44.51 | 17.10 | 44.21 | 22.11 | 48.94 | 14.58 | 13.63 | 18.38 | 11.27 | 34.94 | 12.84 | 18.18 | 27.62 |
| MeanCosine | **53.18** | 33.51 | 44.62 | 19.42 | **44.62** | 23.56 | 47.97 | 13.56 | **17.76** | 20.54 | **14.35** | 31.66 | 13.35 | 18.76 | 28.35 |
| NQR | 45.24 | 24.65 | 39.93 | 14.65 | 35.57 | 19.20 | 44.64 | 11.75 | 13.20 | 13.67 | 10.96 | 26.11 | 11.23 | 16.13 | 23.35 |
| SCORE | 52.79 | **33.59** | **45.93** | **20.69** | 44.31 | **24.23** | **50.38** | **15.96** | 17.27 | **23.53** | 14.23 | **37.55** | **14.41** | **19.52** | **29.60** |
| | | | | | | | NELL995 | | | | | | | | |
| Unconstrained | 47.02 | 23.36 | **41.79** | 29.45 | 33.53 | **22.78** | 47.10 | 23.35 | 21.30 | 29.71 | 15.70 | 31.72 | 10.34 | 29.79 | 29.07 |
| MeanCosine | 46.98 | 22.99 | 41.74 | 28.85 | 32.81 | 21.74 | 47.36 | 22.64 | 22.23 | 27.67 | 14.60 | 28.97 | 10.17 | 29.35 | 28.43 |
| NQR | 37.55 | 14.81 | 33.77 | 20.28 | 26.46 | 13.23 | 37.64 | 14.15 | 15.04 | 21.62 | 7.94 | 23.04 | 6.25 | 21.19 | 20.93 |
| SCORE | **47.58** | **23.99** | 41.74 | **30.43** | **33.63** | 22.59 | **47.57** | **24.81** | **22.98** | **31.32** | **16.54** | **33.38** | **11.02** | **30.51** | **29.86** |

Table 5: NDCG@10 results **target-variable SimCQA** averaged over preference sets of size 10.

| Method | 1ps0 | 2ins0 | 2is0 | 2ps0 | 2ps1 | 2us0 | 3ins0 | 3is0 | 3ps0 | 3ps1 | 3ps2 | inps0 | inps1 | ips0 | ips1 | pins0 | pins1 | pis0 | pis1 | pnis1 | ups0 | ups1 | Avg. |
|---|---|---|---|---|---|---|---|---|---|---|---|---|---|---|---|---|---|---|---|---|---|---|---|
| | | | | | | | | | FB15k237 | | | | | | | | | | | | | | |
| Unconstrained | 43.90 | 23.34 | 47.40 | 29.29 | 28.75 | 34.37 | 37.14 | 50.54 | 28.60 | 27.72 | 27.16 | 22.15 | 21.48 | 40.29 | 34.32 | 21.48 | 20.05 | 36.97 | 41.32 | 9.87 | 30.34 | 27.98 | 31.11 |
| SCORE | 44.43 | 24.07 | 48.06 | 29.39 | 29.84 | 34.91 | 37.80 | 51.38 | 29.00 | 28.36 | 28.34 | 22.65 | 22.62 | 40.42 | 35.61 | 22.36 | 21.20 | 38.12 | 42.03 | 10.57 | 30.68 | 28.73 | 31.84 |
| | | | | | | | | | Hetionet | | | | | | | | | | | | | | |
| Unconstrained | **52.95** | 33.11 | 44.51 | 17.14 | 17.10 | 44.21 | 22.11 | 48.94 | 32.02 | 14.59 | 14.58 | 13.52 | 13.63 | 21.42 | 18.38 | 13.84 | 11.27 | 30.27 | 34.94 | 12.84 | 17.18 | 18.18 | 24.85 |
| SCORE | 52.79 | 33.59 | 45.93 | 21.22 | 20.69 | 44.31 | 24.23 | 50.38 | 32.30 | 20.55 | 15.96 | 16.62 | 17.27 | 26.58 | 23.53 | 17.16 | 14.23 | 30.31 | 37.55 | 14.41 | 18.59 | 19.52 | 27.17 |
| | | | | | | | | | NELL995 | | | | | | | | | | | | | | |
| Unconstrained | 47.02 | 23.36 | **41.79** | 30.98 | 29.45 | 33.53 | **22.78** | 47.10 | 25.64 | 23.55 | 23.35 | 23.26 | 21.30 | 37.64 | 29.71 | 25.20 | 15.70 | 38.29 | 31.72 | 10.34 | 27.57 | 29.79 | 29.05 |
| SCORE | 47.58 | 23.99 | 41.74 | 31.45 | 30.43 | 33.63 | 22.59 | 47.57 | 26.78 | 25.07 | 24.81 | 24.52 | 22.98 | 39.30 | 31.32 | 26.21 | 16.54 | 40.30 | 33.38 | 11.02 | 27.63 | 30.51 | 29.97 |

Table 6: NDCG@10 results for **general SimCQA** averaged over preference sets of size 10.

Following our main experiments, we apply a similarity constraint to a query by using preference sets of size $t = 1, \ldots, 10$. For each value of $t$ we then compute NDCG@10 and report the average over the 10 sizes. We present results for target-variable SimCQA in Table 5 and for general SimCQA in Table 6.

For general SimCQA, we suffix the query type with s$\ell$, where $\ell$ is the zero-based index of the variable on which the similarity constraint is applied. Variables are ordered in a topological order from the leaf nodes to the root of the query graph (see Figure 6). For example, for a 2p chain, the intermediate variable corresponds to index 0 (2ps0), and the terminal variable corresponds to index 1 (2ps1).

## A.7 LLM USAGE

Large language models were used sparingly in preparing this work, limited to assistance with word choice and clarity of exposition.

