# OpenReview forum: "Similarity-Constrained Reweighting for Complex Query Answering on Knowledge Graphs"
_ICLR.cc/2026/Conference — Submitted to ICLR 2026_

### Official Review · Reviewer_htvZ · 2025-10-28

**Soundness:** 3
**Presentation:** 2
**Contribution:** 2
**Rating:** 4
**Confidence:** 4

**Summary:**

This paper proposes the new CQA tasks considering the soft constraints over any variables, which extends the constraints for the free variable in previous works. To address this new setting, this paper proposes a new re-weighting method to interfere with the symbolic search based on the new constraints, where this new method is light and has linear complexity involving two parameters. Then the experiments showed that this new method has advantages over the previous method considering the free variable and the naive baseline.

**Strengths:**

1. Propose a new setting for CQA: consider the additional soft constraints over intermediate variables.
2. Propose a light and effective method to rerank the answer sets based on the new soft constraints and the results are good.

**Weaknesses:**

1. In my opinion, for the paper proposing new tasks, it’s key to introduce the motivation and application of this new setting. After reading this manuscript, I know the soft constraints over intermediate variables are new compared with existing methods.  However, I don’t know why to extend this new setting. In real situation, it's hard to prepare such preference set for re-weighting.
2. The presentation of method is not clear. Though this manuscript provide the details of the re-weighting operations, it's lacking for how this operation integrated with the  symbolic search process.  Is the re-weighting  operation applied over each updating for fuzzy vectors?
3. The compared baselines are limited and all symbolic search methods. It’s important to include more baselines for new task. Query embeddings methods are mainstreams lines for complex query answering, thus I suggest authors can include some classic methods in query embedding methods.

**Questions:**

Typos:
I found some potential typos in this manuscript and listed them in the following:
1.Equation 1 in Line 147, this formula lacks the existential qualifier \exists based on the semantics of the natural language question.
2.Equation 3 in Line 161: the symbolic \exist v_{\lneg i} \in \mathcal{E} only is right for two variables. Please consider the general formula for any variables.
3.Line 69: e as a symptom,” a user

---

> ### Author Response · Authors · 2025-11-15
>
> Thank you for your thoughtful review and your constructive feedback. Your comments helped us refine both the motivation and the methodological clarity of the revision (changes shown in green).
>
> **Task motivation:** Our goal is to expand CQA beyond purely logical constraints. In many practical scenarios, such as retrieving entities “similar to” a given example, these preferences cannot be captured by first-order logic alone. We agree that in the real world, preference sets might be hard to prepare, which is why we explicitly focus our evaluation on small sets of maximum 10 instances.
>
> **On the presentation of the method:** Thank you for noting this. We have added an additional figure (Fig. 2) and clarified how reweighting integrates with symbolic projection and intersection. This makes the interaction between SCORE updates and the fuzzy vector propagation more explicit.
>
> **Query embedding methods:** Our approach is designed to extend neuro-symbolic CQA methods that explicitly propagate fuzzy vectors through the query structure. These methods naturally expose intermediate scores for each variable and operator, which allows similarity constraints to be injected at specific points in the query. In contrast, embedding-based methods embed the entire query into a single vector, making it non-trivial to incorporate similarity constraints on arbitrary variables. Investigating how to adapt such methods is an interesting direction for future work, which we now discuss in Section 7.
>
> **On the formulas:** Thank you for the keen observations! For brevity, in Equation 1 we refer to the logical formula only comprised by predicates and connectives. We leave the use of the existential quantifier to answer sets (as in Eq. 3). In this formula, we overload the existential quantifier notation to several variables: we define (Line 159) $\boldsymbol{v}_{\neg i}$ as the set of all variables minus $v_i$, to which we apply the existential quantifier.

---

> > ### Comment · Reviewer_htvZ · 2025-11-27
> >
> > Thank you for the reply. However, the concerns of the presentation and motivation remain. I am willing to keep my scores.

---

> > > ### Author Response · Authors · 2025-11-27
> > >
> > > Thank you again for your detailed review and for the helpful suggestions. If there is a specific step in the motivation or the new figure explaining the integration with symbolic search that still seems unclear, we would be grateful for any brief indication that could help us further improve the camera-ready version, if you have some time.
> > >
> > > Thank you once again for your thoughtful comments and for helping us strengthen the paper.

---

### Official Review · Reviewer_MS4r · 2025-10-31

**Soundness:** 2
**Presentation:** 2
**Contribution:** 2
**Rating:** 4
**Confidence:** 4

**Summary:**

This paper proposes a new method for complex query answering. Unlike traditional settings, it focuses on scenarios where similarity constraints are applied to either intermediate nodes or answer nodes within complex queries. The key idea is to represent the potential answers of each variable as fuzzy sets, and then perform Similarity-Constrained Reweighting to adjust these fuzzy vectors accordingly. Overall, the method is conceptually simple yet addresses an important and underexplored problem.

**Strengths:**

The paper is clearly written and easy to follow. The writing quality is excellent, and the authors address a novel and meaningful problem—introducing similarity constraints on both intermediate variable nodes and answer nodes in complex query answering.

The inclusion of theoretical analysis enhances the soundness and credibility of the proposed algorithm, while the experimental results convincingly demonstrate its effectiveness across different datasets and settings.

**Weaknesses:**

The core idea of the paper is simple and intuitive, with the primary contribution being the introduction of the Similarity-Constrained Reweighting mechanism. However, this contribution appears somewhat limited in scope.

The experimental section requires significant improvement for better clarity and completeness. First, although the paper introduces a new problem setting, it does not provide sufficient details on how existing methods (such as SCORE, NQR, and other baselines) were adapted to this new domain. This information is essential and should be described explicitly.

Second, the presentation of experiments is not very clear. Both the experimental setup and the performance analysis are difficult to follow. The section would benefit from a thorough revision to improve organization, explanation, and readability.

**Questions:**

no

---

> ### Author Response · Authors · 2025-11-15
>
> Thank you for your thorough and constructive review. Your feedback has helped us improve our revised version of the paper (changes in the PDF are shown in green).
>
> **On the scope:** We would like to highlight that SCORE has a broad scope of implications: our experiments show that it extends the types of queries we can answer with methods for complex query answering, which so far had been limited to logical constraints; it is a lightweight (only two parameters need to be tuned) and interpretable method; and it is applicable to the majority of existing methods for complex query answering. We now discuss in a newly added Discussion section (section 7 in our updated revision).
>
> **On clarity of new problem setting:** Thank you for this suggestion. We have added Figure 2, as well as clarifications in the text describing how we adapt the methods. We hope this can clarify how methods are adapted to the new problem setting.
>
> **Presentation of experiments:** We have tried our best at structuring a clear experiments section, with further details in the Appendix when space was a limitation. In case there are specific sections or terms in the analysis that are hard to follow, could you please specify which ones? This could help us improve the presentation of our results.

---

### Official Review · Reviewer_3aWr · 2025-10-31

**Soundness:** 2
**Presentation:** 3
**Contribution:** 2
**Rating:** 4
**Confidence:** 4

**Summary:**

This paper proposes a new extension to complex query answering (CQA) over knowledge graphs (KGs), the paper names it similarity-based constraints CQA. The similarity constraint can be applied to any variable within this query. To address this new question, the paper proposes SCORE (Similarity-COnstrained REweighting). It achieves this via a logit-space reweighting mechanism that only contains two new hyperparameters.

**Strengths:**

1.	It is praiseworthy to introduce the novel generalization of similarity constraints.
Notably, the extension of SimCQA allows for similarity constraints of intermediate variables. Overall, the paper addresses a realistic but previously unstudied case in CQA.
2.	SCORE has good interpretability. Unlike black-box neural methods, SCORE’s update mechanism is transparent and traceable to individual preference contributions.
3.	The methodology of SCORE is generally easy to comprehend and follows, like its logit-space reweighting mechanism, the paper also proposes some theoretical results to show its soundness.
4.	The code, datasets are provided.

**Weaknesses:**

1.	The setting of similarity function is far too simple. The paper just uses binary classification of True/False to determine whether one entity is similar to another one in the problem setting. This is quite simple but another problem is bigger: the paper does not explain very clearly how it decides the ground truth of similarity. To my understanding, the paper uses other answers from same query as ground truth which can be very problematic.

2.	The performance is highly dependent on the backbone model as it only introduces two new hyperparameters. I also found that the experimental performance falls behind NQR in pairwise accuracy and also only outperforms very simple baselines like MeanCosine slightly. Therefore, I suspect the effectiveness of SCORE.

**Questions:**

Perhaps using numerical attribute (which is provided in NELL and other dataset) instead of Boolean similar/not similar is a better alternative. Have you considered that?

---

> ### Author Response · Authors · 2025-11-15
>
> Thank you for your thoughtful and constructive review! Your comments have substantially improved the clarity of our revision (changes shown in green).
>
> **Determining similarity:** Due to space constraints, we describe how the ground truth of similarity is determined in Appendix A.3: following prior work, we partition answers into disjoint clusters of embeddings that encode textual semantic similarity. To avoid any problematic issues we use only answers that are reachable by graph traversal (i.e. following the training edges), while the actual evaluation is done on answers that are not reachable by edges in the training set.
>
> **Performance vs NQR and MeanCosine:** Pairwise accuracy (PA) measures whether preferred answers outrank non-preferred ones, but it does not account for global ranking quality. Methods such as NQR maximize PA aggressively, often at the cost of degrading overall query-answering performance (MRR, H@10, NDCG@10). This is consistent with observations in prior learning-to-rank work. SCORE achieves a more balanced tradeoff across these metrics, and unlike MeanCosine, it also supports similarity constraints on arbitrary variables.
>
> **Using numerical attributes:** We agree this is an interesting direction. Textual descriptions already capture some numerical information implicitly, but incorporating explicit numerical attribute similarity could enrich the constraint signal. We now discuss this in the revised Discussion section (Section 7).

---

> > ### Comment · Reviewer_3aWr · 2025-11-26
> >
> > Thanks for the explanation, yet in Appendix A.3 you point out that the ground truth of similarity just comes from the embedding rather than human annotation, therefore not a reliable task.

---

> > > ### Author Response · Authors · 2025-11-26
> > >
> > > We thank the reviewer for raising this concern. We agree that human-annotated similarity judgments represent the most direct notion of “true” semantic similarity. However, relying exclusively on human studies would make it very difficult to conduct large-scale evaluations at the scale of our work, comprising thousands of queries on several KGs, which is precisely why we adopt embedding-based similarity as a computational proxy.
> > >
> > > Importantly, prior work has shown that distances in embedding spaces correlate with human perceptions of semantic similarity and relatedness (e.g., Reimers & Gurevych, 2019; Grand et al., 2022; Merkx et al., 2022). While we acknowledge that embedding-derived clusters are not a perfect substitute for human ground truth, they provide a systematic, reproducible, and scalable way to approximate semantic groupings, enabling us to analyze preference-based reranking behavior across a broad range of query types. In our latest version, we have now expanded Appendix A3 to further clarify this important point.
> > >
> > > **References**
> > >
> > > Reimers, N. & Gurevych, I. (2019). Sentence-BERT. EMNLP-IJCNLP.
> > >
> > > Grand, G. et al. (2022). Semantic projection from word embeddings. Nature Human Behaviour.
> > >
> > > Merkx, D. et al. (2022). Visually grounded word embeddings. Proc. CMCL Workshop (ACL).

---

### Official Review · Reviewer_HSJ3 · 2025-11-01

**Soundness:** 4
**Presentation:** 4
**Contribution:** 4
**Rating:** 8
**Confidence:** 4

**Summary:**

The paper introduces a method for complex query answering (CQA) on knowledge graphs, where entity similarity constraints are exploited to guide the solver towards solutions that are consistent with such constraints. In particular, the method extends a previous one in order to exploit an entity similarity constraint defined on arbitrary variables, rather than on only the answer entity (also called target). While the method is rather simple, it turns out that it can achieve a better CQA accuracy with little computational overhead.

**Strengths:**

I have found the writing and the plots to be extremely clear. The writing slowly introduces concepts when they are needed with many examples and a sufficiently detailed notation. Overall, I have also appreciated the simplicity of the proposed method, as well as the execution of the experiments. From a quick look, it looks like the code can help reproducing all the presented results.

**Weaknesses:**

Although the CQA benchmarks used in the paper are particularly established in the community, some queries can suffer from link leakage from the training set. For instance, most of 2p complex queries can actually be decomposed into the simplest task--link prediction (1p queries)--if one considers also the training triples at test time. This means that ranking metrics for certain query types are actually inflated. Recently, [A] showed this problem and proposed a new set of much more challenging CQA benchmarks, where all triples in a query are missing in the observed knowledge graph and therefore must be predicted. Evaluating the baselines and the proposed method on these other recent benchmarks could strengthen the value of the obtained empirical results. Although I do not expect a huge difference w.r.t. the presented results, I suggest the authors to evaluate their method on these other datasets as well.

[A] C. Gregucci, B. Xiong, D. Hernández, L. Loconte, P. Minervini, S. Staab, A. Vergari. Is Complex Query Answering Really Complex? ICML 2025.

**Questions:**

- The definition of similarity-constrained complex queries assumes that there exists a single similarity constraint for one of free variables (e.g., see Eq. 6). Do you think this framework can be further extended to a setting where several of the free variables take part in a similarity constraint?

- What is the "unconstrained" baseline reported in Figure 2? Is it evaluated on the same dataset for SimCQA? I do not understand why the unconstrained baseline performs better than the baseline NQR, which instead takes into account similarity constraints. Could you please clarify this aspect?

---

> ### Author Response · Authors · 2025-11-15
>
> Thank you for your thoughtful and constructive review! Your comments have helped us significantly improve the revised version (changes shown in green).
>
> **On evaluating the new benchmarks:** We appreciate the suggestion to evaluate on the benchmarks of Gregucci et al. (2025). Extending our results to settings where all query triples are missing is indeed interesting. However, adapting our pipeline requires substantial preprocessing (e.g., reconstructing query sets with similarity constraints) and compute. We have begun exploring this process and will report insights when feasible. We agree that evaluating SCORE in this setting is an exciting future direction.
>
> **Similarity constraints on several variables:** Thank you for raising this. Since SCORE applies reweighting independently to the fuzzy vector of each variable, it can naturally be applied to multiple variables without modifying the mechanism. The main challenge becomes experimental: evaluating all combinations of constrained variables leads to a combinatorial search space. We now discuss this point in the revised Discussion section (Section 7).
>
> **Unconstrained:** “Unconstrained” refers to the base QA model (QTO in our experiments, we now make this more explicit in Section 6.1.), which is not designed to incorporate similarity constraints. We include this baseline to contextualize the performance of constraint-aware methods. While NQR uses similarity constraints, it tends to over-optimize pairwise accuracy (PA) at the expense of overall ranking metrics such as MRR, H@10, and NDCG@10.

---

> > ### Comment · Reviewer_HSJ3 · 2025-11-26
> >
> > The authors addressed all my questions. I will keep my score as is.

---

### Author Response · Authors · 2025-12-02
**Summary of discussion**

Dear AC, reviewers,

We thank you for your careful assessments. Below we provide a concise summary of the consensus, and the precise changes made in the revised version.

Across reviewers, there is agreement that:

- The submission is novel and addresses a previously unexplored extension of CQA, namely applying similarity constraints to *arbitrary variables*, not only the target (3aWr, MS4r, htvZ).
- The method is conceptually simple, computationally lightweight, and interpretable (HSJ3, 3aWr, MS4r, htvZ).
- The paper is clearly written, with strong notation, examples, and well-executed experiments (HSJ3; MS4r: “writing quality is excellent”).
- Most concerns relate to presentation, clarity of motivation, or additional discussion.

We provided answers to all questions by the reviewers, whose feedback helped us improve our revised version of the paper as follows (changes in the PDF are shown in green color):

1. **New Discussion section (Section 7).**
   Clarifies implications for multi-variable constraints (HSJ3), the role of numerical attributes (3aWr), and why query-embedding baselines cannot directly support arbitrary-variable constraints (htvZ).
2. **New figure and clearer integration with symbolic search.**
   Added Figure 2 and explanatory text to show how SCORE interacts with projection/intersection steps, addressing MS4r and htvZ.
3. **Expanded Appendix A.3 on modeling similarity.** We clarify that similarity sets derive from embedding-based clusters of textual descriptions, following prior work indicating that embedding distances correlate with human similarity judgments (addressing 3aWr).
4. **Additional baseline descriptions and discussion of results.**
   We clarify the “Unconstrained” baseline (requested by HSJ3), explain how each baseline incorporates constraints, and discuss the trade-off between pairwise accuracy and global rankings (addressing HSJ3, 3aWr).

---

### Meta-Review · Area_Chair_prxw · 2026-01-06

**Summary:**

The paper introduces the task of similarity-constrained complex query answering (simCQA) where, given a knowledge graph, one has to answer a query (as in vanilla CQA) but subject to additional soft-constraints about the similarity of certain intermediate entities to some given preferences. Then, a simple baseline for simCQA is provided, SCORE, which adapts classical fuzzy-based variants for CQA to simCQA by reweighting their scores by a similarity after normalization to keep them as pseudo-probability values in [0, 1].

Reviewers agreed that this is an interesting direction and found that the paper is generally well-written and easy to follow.
I agree that simCQA is indeed interesting, but I also note that in practice is a bit incremental when compared to [1]. While it can still be an important generalization, this needs to be backed up by more realistic experiments, while the authors mostly reuse the synthetic setting of [1]. As such, I agree with reviewer htvZ.

Other concerns found by reviewers are the lack of details on the methodology, e.g., how to integrate SCORE in the search process (answered by the authors in the rebuttal) and a comparison w.r.t. other (more sophisticated) alternatives to CQA that do indeed try to renormalize scores for different predicates. The authors claim that SCORE is more interpretable, but I believe monotonicity is not enough to support this claim. Furthermore, their renormalization scheme is heuristic and does not guarantee their interpreted probabilities are calibrated (an experiment could be easily conducted to check for this). Lastly, as in principle any fuzzy-based CQA could be used as a baseline, how would they perform?

I am a bit torn on this paper, as I recognize some potential, but it stands it is borderline. I think it has merit and I invite authors to make an additional round of revision, including ablations and a strong motivation.

[1] Daniel Daza, Alberto Bernardi, Luca Costabello, Christophe Gueret, Masoud Mansoury, Michael Cochez, and Martijn Schut. Interactive query answering on knowledge graphs with soft entity constraints. arXiv preprint arXiv:2508.13663, 2025

**Reviewer Concerns:**

HSJ3 highlighted how current benchmarks for CQA can be flawed and suggested that also the ones for simCQA could suffer from this issue, the authors agreed in principle but did not carry additional experiments. The reviewer seemed not bothered by it. For me this remains an important potential issue to rule out.

3aWr would still be concerned about the method being too "simple". I do not agree with their argument against simplicity of a solution (simplicity can be elegant), but I agree that there is some incrementality behind the work and not enough ablations.

MS4r criticised (what I understand) as lack of details about how to turn any (fuzzy) CQA method in a simCQA method with SCORE and how this would affect performance. Only a small gridsearch over hyperparameters is provided and it is not clear what is the methods' sensitivity to them.

The strongest left concern for htvZ is the motivation behind this new task. I agree with this point (see above), and I did not find the short answer by the authors to be convincing (they could have listed some concrete real-world scenarios where this is actually already happening).

**Reviewer Scores:**

HSJ3 was the most enthusiastic for the paper, giving 8, but I believe they would have lowered the score to 6 after discussion with other reviewers.

3aWr would have likely kept their score of 4 likely, with some little probability increasing to 6.

MS4r would likely keep the score to 4 as well as htvZ as their core concerns are left unanswered.

---

### Decision · Program_Chairs · 2026-01-26

Reject